# Establishment of an early diagnosis model of colon cancerous bowel obstruction based on 1H NMR

Jie Zeng[1], Jin Peng[2]*, Hua Jiang[1], Pengchi Deng[3], Kexun Li[1], Daolin Long[1], Kai Wang[1]

1 Department of Emergency Surgery, Sichuan Academy of Medical Sciences & Sichuan Provincial People's Hospital, Chengdu, Sichuan, P.R. China, 2 Department of Histology Embryology and Neurobiology, Sichuan University West China School of Basic Medical Sciences and Forensic Medicine, Chengdu, Sichuan, P.R. China, 3 Analytical and Testing Center, Sichuan University, Chengdu, Sichuan, P.R. China

* pengjin@scu.edu.cn

## Abstract

### Objective

To prospectively establish an early diagnosis model of acute colon cancerous bowel obstruction by applying nuclear magnetic resonance hydrogen spectroscopy(1H NMR) technology based metabolomics methods, combined with machine learning.

### Methods

In this study, serum samples of 71 patients with acute bowel obstruction requiring emergency surgery who were admitted to the Emergency Department of Sichuan Provincial People's Hospital from December 2018 to November 2020 were collected within 2 hours after admission, and NMR spectroscopy data was taken after pretreatment. After postoperative pathological confirmation, they were divided into colon cancerous bowel obstruction (CBO) group and adhesive bowel obstruction (ABO) control group. Used MestReNova software to extract the two sets of spectra bins, and used the MetaboAnalyst5.0 website to perform partial least square discrimination (PLS-DA), combining the human metabolome database (HMDB) and the Kyoto Encyclopedia of Genes and Genomes (KEGG) to find possible different Metabolites and related metabolic pathways.

### Results

22 patients were classified as CBO group and 30 were classified as ABO control group. Compared with ABO group, the level of Xanthurenic acid, 3-Hydroxyanthranilic acid, Gentisic acid, Salicyluric acid, Ferulic acid, Kynurenic acid, CDP, Mandelic acid, NADPH, FAD, Phenylpyruvate, Allyl isothiocyanate, and Vanillylmandelic acid increased in the CBO group; while the lecel of L-Tryptophan and Bilirubin decreased. There were significant differences between two groups in the tryptophan metabolism, tyrosine metabolism, glutathione metabolism, phenylalanine metabolism and synthesis pathways of phenylalanine, tyrosine and tryptophan (all P<0.05). Tryptophan metabolism pathway had the greatest impact

**Data Availability Statement:** All relevant data are within the manuscript and its Supporting information files and the raw data can be download at https://github.com/dcpengjin/metabolomics_data.

**Funding:** The study was supported by Basic scientific research business expenses of Science & Technology Department of Sichuan Province (grant no.2018YSKY0017-9), and the funders had no role in study design, data collection and analysis, decision to publish, or preparation of the manuscript. Kexun Li and Daolin Long who were studying for a master's degree got ¥12,000 as Labor wage from this foundation, respectively. Mean while, the other authors of this manuscript got no salary from the funder.

**Competing interests:** The authors have declared that no competing interests exist.

**Abbreviations:** 1H-NMR, nuclear magnetic resonance hydrogen spectroscopy; 3-HAA, 3-hydroxyanthranilic acid; ABO, adhesive bowel obstruction; AhR, aryl hydrocarbon receptor; AITC, allyl isothiocyanate; BILI, bilirubin; BMI, body mass index; CBO, colon cancerous bowel obstruction; CEA, carcinoembryonic antigen; DSPC, debiased sparse partial correlation; FA, ferulic acid; GA, gentisic acid; Glu, glutamate; GluR, Glu receptor; HMDB, the human metabolome database; HMOX-1, heme oxygenase-1; KA, Kynurenic acid; KEGG, the kyoto encyclopedia of genes and genomes; KP, kynurenine pathway; Kyns, kynurenines; MA, mandelic acid; NRs, N-methyl-D-aspartic acid receptors; PCA, principal component analysis; PLS-DA, partial least squares discrimination; PPA, phenylpyruvate; SUA, salicyluric acid; Try, tryptophan; VIP, variable important in projection; VMA, vanillylmandelic acid; XA, Xanthurenic acid.

(Impact = 0.19). The early diagnosis model of colon cancerous bowel was established based on the levels of six metabolites: Xanthurenic acid, 3-Hydroxyanthranilic acid, Gentisic acid, Salicylic acid, Ferulic acid and Kynurenic acid ($R2 = 0.995$, $Q2 = 0.931$, RMSE = 0.239, AUC = 0.962).

## Conclusion

This study firstly used serum to determine the difference in metabolome between patients with colon cancerous bowel obstruction and those with adhesive bowel obstruction. The study found that the metabolic information carried by the serum was sufficient to discriminate the two groups of patients and provided the theoretical supporting for the future using of the more convenient sample for the differential diagnosis of patients with colon cancerous bowel obstruction. Quantitative experiments on a large number of samples were still needed in the future.

## Introduction

Due to changes in dietary patterns, obesity, and the prevalence of smoking, the incidence of colorectal cancer in East Asia and other areas where the incidence of colorectal cancer was low in the past is increasing rapidly [1]. At the same time, due to traditional reasons such as awareness of medical care, a large number of colorectal cancer patients in China are already in the advanced stages when they are diagnosed. Colon cancerous bowel obstruction (CBO) is a frequent end-stage event in patients with advanced colon cancer, and its treatment effect is also extremely pessimistic. This is prominently reflected in the contradiction between the uncertainty of preoperative diagnosis and the urgency of surgical intervention, the wide surgical indications and the high difficulty of operation, as well as the poor prognosis of treatment and the high expectations of patients [2]. Although NCCLS established the diagnostic criteria for Malignant bowel obstruction (MBO) in 2007 [3], it has been found that its application is limited by the dependence on imaging evidence and the low specificity of tumor-associated antigens in clinical practice. Patients with CBO have experienced a systemic metabolic disorder that is regulated by hormones, proteins, and small molecule chemical groups produced from metabolism. If we could understand and depict the complete metabolome of patients with CBO, It's possible to quickly assess the patient's condition and predict its prognosis. In the past ten years, the research on the metabolic profile of patients with colorectal cancer has mostly focused on the early diagnosis and staging of tumors. It was rare to describe the metabolic characteristics of colon cancerous patients with bowel obstruction. This research is based on 1H-NMR, using chemometrics and mathematical methods to construct a diagnostic model for CBO patients for the first time. It's hoped that the clinical goals of accurate diagnosis and precise treatment could be achieved through the establishment of the metabolic profile of CBO.

## Materials and methods

### Research object

The emergency patients who attended the Emergency Surgery Department of Sichuan Provincial People's Hospital from December 2018 to November 2020 were enrolled. The research was approved by the ethics review committee of Sichuan Provincial People's Hospital (NO. 2018[233]), and all patients signed the written informed consent before participating in the study.

1. Inclusion criteria: (1) Older than 18 years; (2) Hospital admission for acute abdomen; (3) Positive clinical evidences of bowel obstruction [including medical history, physical examination and imaging evidences]; (4) Negative digital anal examination; (5) Fasting for more than 24 hours; (6) Positive emergency exploratory Indications (whole abdomen Tenderness and rebound pain; bowelsound <1 per min).

2 Exclusion criteria (1) Combined with hyperthyroidism, diabetes and other metabolic diseases, or accompanied by neurological diseases that affect metabolism; (2) For patients with abnormal liver and kidney function; (3) Long-term administration of hypoglycemia/antihyperlipidemic agents, thyroxine tablets or other drugs that affect metabolism; (4) Sepsis; (5) Pregnant women; (6) Patient got medical procedures (e.g., oral or intravenous administration, gastrointestinal decompression, enemas) within 48 hours; (7) Patient got imaging examination using contrast agents within 48 hours; (8) Patient got intravenous or oral rehydration prior to specimen collection.

3. Sample size According to the inclusion and exclusion criteria above, the study included 71 patients with acute bowel obstruction. Then 7 cases of severe metabolic abnormalities were discharged (5 cases of metabolic disease that were diagnosed for the first time, 2 cases of severe abdominal infection); 7 cases of other neoplastic bowel obstruction (3 cases of intestinal lymphoma, 4 cases of extraintestinal malignant lesions invasion) were discharged; 5 cases were discharged without postoperative pathological diagnosis (2 cases refused surgery, 3 cases could not obtain pathological results). 22 patients were included in the colon cancerous bowel obstruction group (CBO), while 30 patients were in the adhesive bowel obstruction group (ABO).

## Experimental method

**1 Patient information and specimen collection.** *1.1 General information and collection of pathological results*. The postoperative paraffin pathological results were checked 1 week after the operation and the patients who met the enrollment criteria were registered. The general information included name, gender, age, Body Mass Index(BMI), carcinoembryonic antigen (CEA) and preoperative abdominal CT.

*1.2 Specimen collection*. Collected 3ml whole blood into vacuum blood collection tubes (Blue cap with sodium citrate, 10.25mm×64mm, batch number 363095, American BD company) from the peripheral veins of all participants within 2 hours after admission, These samples were all processed within 30 minutes. The tube was centrifuged at 16000r/min for 10 minutes, then the supernatant was taken and transferred to EP tube, which was refrigerated at -80°C finally.

**2 Sample processing.** Placed—80°C sample to room temperature firstly. 60μl D2O was added into an EP tube with 450μl sample, and vortexed for 30 seconeds in a vortex machine. Then the tube was transferred to a 5mm Wilmad NMR tube for computer testing and analysis lastly.

**3 Sample testing.** A Bruker 600MHz NMR spectrometer (DRX 600MHz NMR, Bruker Biospin Rheinstetten, Germany) was used to detect the NMR tube, and all one-dimensional 1H-NMR spectra were obtained. The proton frequency was set to 600.1 MHz and a 300K cryogenic probe was used. The 1H-NMR experiment used the Carr-Purcell-Meiboom-Gill (CPMG) pulse sequence to obtain one-dimensional NMR spectra from each sample. The spectrum was collected when the spectral width was 20 ppm and the relaxation time was 5 seconds.

**4 Data export.** Used the mestRenova software (version 12.0.0, MestreLab Research, Spain) to open all the spectra that need to be processed and generated a new overlapped spectrum. The Fourier transform is completed after manual phasing. After normalization, the

integration range was set to 0.04-8ppm, the integration interval was 0.04ppm, the spectrum area was divided into 199 intervals of equal width, and the spectral bins were obtained.

**5 Data analysis.** Firstly, used MetaboAnalyst 5.0 to perform partial least squares discrimination (PLS-DA) on the spectral bins, and obtained interval ranking and principal components analysis model with Variable important in projection (VIP)>1, then perform cross validation. Secondly, confirmed the different metabolites by using the Human Metabolome Database (HMDB), combining published research articles and comparing with standard data. Then, analyzed the the peak intensities to evaluate the reliability of the different metabolites between the two groups. Combined with Kyoto Encyclopedia of Genes and Genomes (KEGG), MetaboAnalyst 5.0 was used to analyze different metabolites and their potential metabolic pathways lastly.

**6 Statistical methods.** Chi-square test (Fisher's exact probability test) and t-test (Student's t-test or Wilcox test) were used to analyze categorical variables and continuous variables, and p <0.05 was considered to have a significant difference. Partial least squares discriminant analysis (PLS-DA) was used for multivariate analysis of metabolic differences between the two groups. Debiased Sparse Partial Correlation (DSPC) was used to perform pairwise correlation analysis of differential metabolites. The AUC of the ROC curve was calculated to judge sensitivity.

## Results and discussion

### Results

**1. Baseline data.** In this study, a total of 71 patients were enrolled, in which 19 abnormal cases were discharged, and a total of 52 valid cases were collected. Among them, 22 were confirmed CBO patients and 30 were ABO patients. General informations are in Table 1. The positive rate of CT (AUC = 0.735, CI: 0.591–0.879, P = 0.004) and the level of CEA (AUC = 0.644, CI: 0.488–0.800, P = .048) of the CBO group are significantly higher than those of the ABO group(all P<0.05).

**2. 1H NMR metabolic spectrum of serum samples.** The 1H NMR spectrum of the two groups of serum samples were overlapped with the mestRenova software after processing. It can be seen that there is no significant difference between two groups (Figs 1 and 2). Set the integration range from 0.04 to 8ppm and the integration interval in 0.04ppm. The spectrum area is divided into 199 intervals of equal width to obtain the spectral bins.

**3. Differential metabolite analysis.** Firstly, Used MetaboAnalyst 5.0 to perform non-supervised principal component analysis (PCA) on the spectral bins, showing that there are significant differences between the CBO group and the ABO group (Fig 3). Driven by this observation, we continued to conduct supervised statistical analysis, namely partial least squares discrimination (PLS-DA), and found that the two groups of metabolites were well

**Table 1. Baseline data.**

|  | CBO | ABO | P |
|---|---|---|---|
| Age (mean±sd years) | 67.773±14.3425 | 60.933±17.1302 | 0.135 |
| Height (mean±sd cm) | 159.727±7.8630 | 162.467±8.4271 | 0.239 |
| Weight (mean±sd Kg) | 54.591±8.1220 | 55.567±10.8824 | 0.725 |
| BMI(mean±sd) | 21.541±2.9730 | 21.04±3.1169 | 0.562 |
| Gender(M/F) | 9/13 | 20/10 | 0.065 |
| CT positive rate (%) | 63.6(14/22) | 16.7(5/30) | 0.001 |
| CEA (mean±sd ng/ml) | 23.7850±16.75588 | 3.8727±3.59272 | 0.021 |

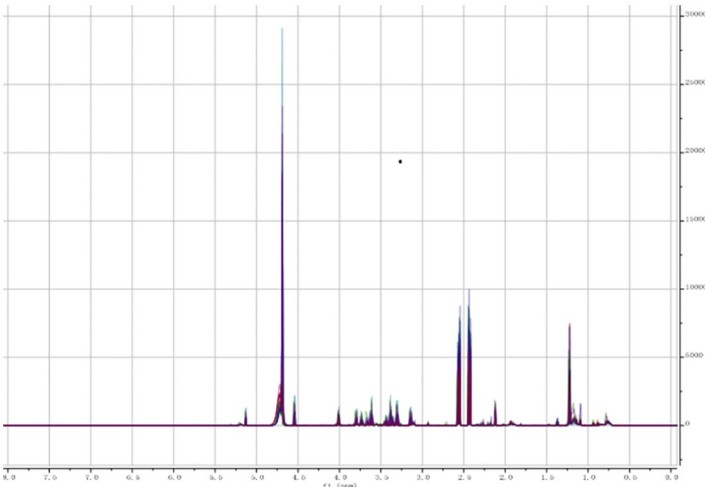

**Fig 1. CBO's spectrums.**

dispersed and have no obvious outliers and specific aggregation trends. Obvious trend of separation between groups is observed (Figs 4 and 5). There were 35 intervals which the Variable Important in Projection (VIP)>1 (Fig 6). They were input into the human metabolome database (HMDB) searching, and 1470 possible metabolites were obtained, of which 388 endogenous metabolites could be detected in the blood. According to the Jaccard Index Match Ratio, combined with published research articles [4–15], the top 30 metabolites were selected. By comparing standard spectra with HMDB (1H NMR Spectrum [1D, 600 MHz, D2O, anticipated]), it was shown that 15 metabolites had the lowest peaks at chemical shifts that overlapped with the above 35 intervals. Mesrenona software was used to determine the peak intensities of these 15 shifts separately, and a t-test revealed that the peak intensities of six shifts were substantially different. Six chemical shifts correspond to six metabolites, and the integrals of the lowest peak intensity are notably different (Table 2 and Fig 7). Compared to the ABO group, the level of Xanthurenic acid (XA), 3-hydroxyanthranilic acid (3-HAA), Gentisic acid

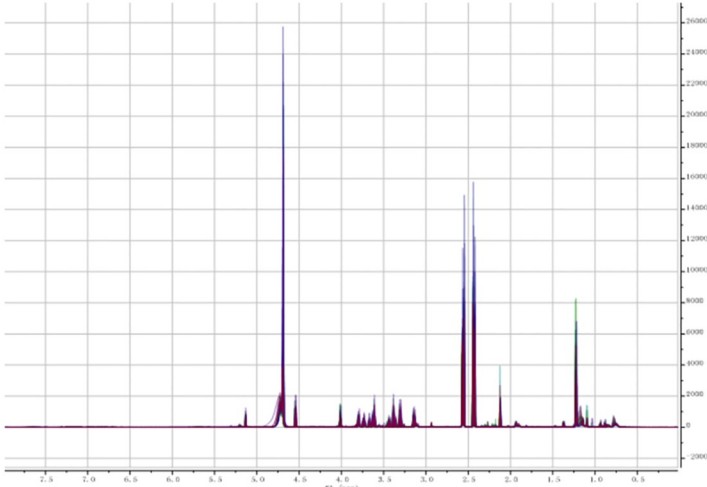

**Fig 2. ABO's spectrums.**

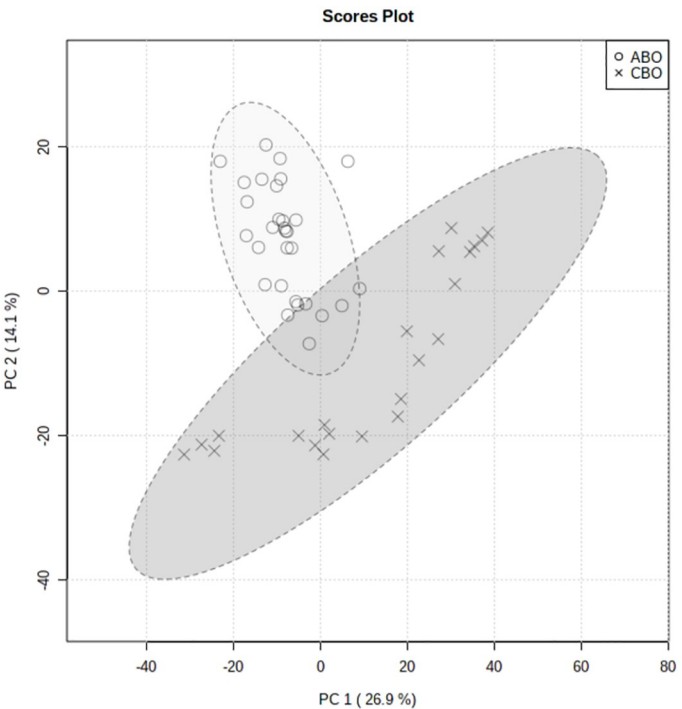

**Fig 3. PCA scores plot between the Comp1 and Comp2.**

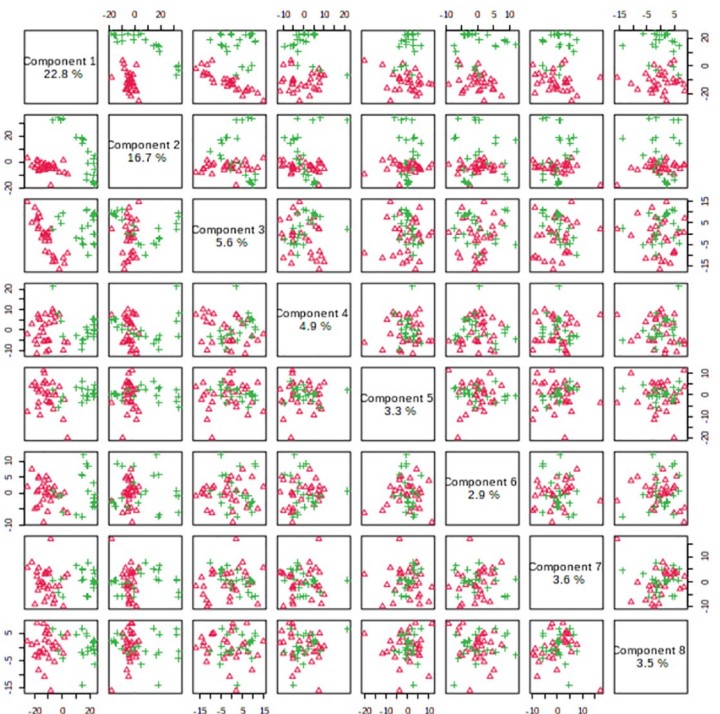

**Fig 4. Pairwise scores plots between the selected components.**

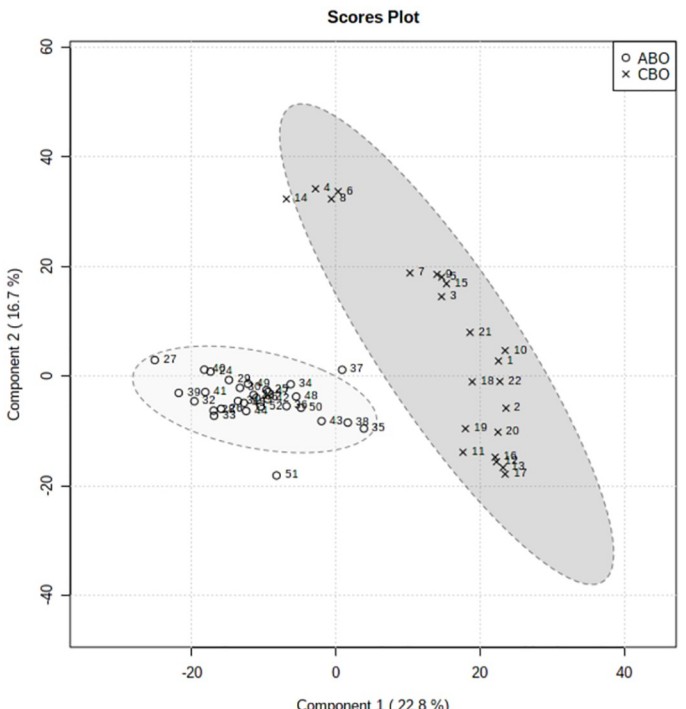

**Fig 5. PLS-DA scores plot between the Comp1 and Comp2.**

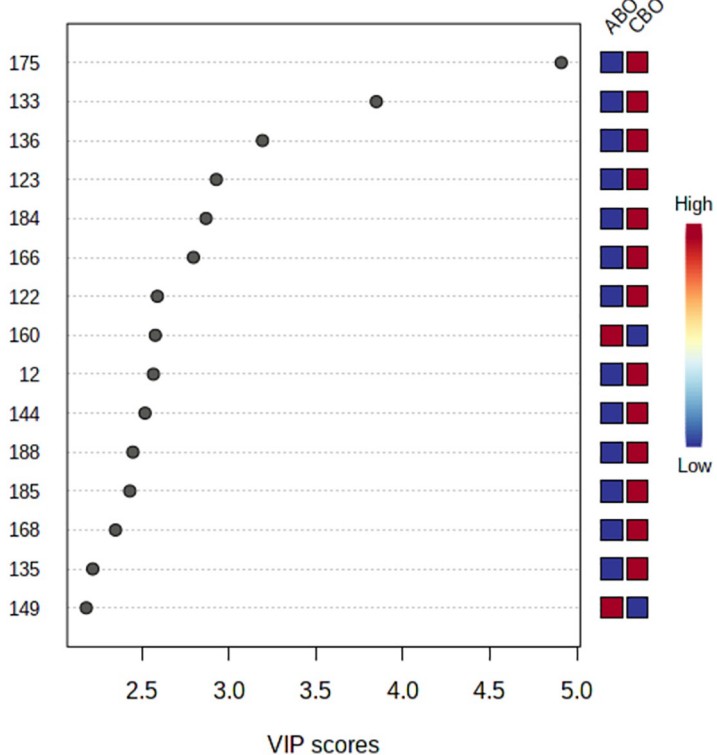

**Fig 6. Important features identified by PLS-DA.**

**Table 2. Comparison of the lowest peak area of metabolites.**

| metabolite | Interval | Jaccard | Minimum peak intensity integral (Mean (SD)) | |
|---|---|---|---|---|
| | | | CBO | ABO |
| Xanthurenic acid | 176 | 0.158 | 907.964 (169.803) | 757.325 (201.528) ↓* |
| Allyl isothiocyanate | 101 | 0.148 | 28634.723 (20289.228) | 24693.512 (16034.627) ↓ |
| Kynurenic acid | 168 | 0.143 | 360.680 (195.294) | 266.103 (156.627) ↓* |
| Mandelic acid | 123 | 0.105 | 814.341 (1205.804) | 734.308 (1664.378) ↓ |
| Gentisic acid | 144 | 0.103 | 4296.669 (2121.122) | 1752.036 (1413.397) ↓* |
| FAD | 166 | 0.103 | 360.680 (195.294) | 266.103 (156.627) ↓ |
| 3-Hydroxyanthranilic acid | 188 | 0.098 | 3502.353 (13572.471) | 68.253 (61.656) ↓* |
| Vanillylmandelic acid | 172 | 0.098 | 895.535 (3099.663) | 210.336 (94.918) ↓ |
| Phenylpyruvic acid | 160 | 0.095 | -4.067 (52.152) | -7.924 (33.538) ↓ |
| NADPH | 101 | 0.092 | 28634.723 (20289.228) | 24693.512 (16034.627) ↓ |
| Ferulic acid | 183 | 0.091 | 601.120 (219.776) | 269.936 (194.209) ↓* |
| Salicyluric acid | 175 | 0.091 | 750.304 (103.508) | 544.265 (154.842) ↓* |
| Bilirubin | 157 | 0.089 | -35.597 (40.246) | -17.707 (41.004) ↑ |
| L-Tryptophan | 196 | 0.089 | 25.432 (98.214) | 784.480 (3647.465) ↑ |
| CDP | 108 | 0.089 | 1126.090 (630.975) | 956.864 (483.452) ↓ |

↓: The peak intensity is lower in ABO group.

↑: The peak intensity is high in ABO group.

*: P<0.005

(GA), Salicyluric acid (SUA), Ferulic acid (FA), Kynurenic acid (KA), CDP, Mandelic acid (MA), NADPH, FAD, phenylpyruvate (PPA), Allyl isothiocyanate (AITC) and Vanillylmandelic acid (VMA) increase relatively; while the level of L-tryptophan (L-Try) and bilirubin (BILI) decrease relatively in the CBO group. Performed K-fold cross validation to prevent overfitting to obtain a discriminant model for CBO by MetaboAnalyst 5.0: Comp1 = 176(interval) *0.228 +188 *0.167+ 144 *0.056+ 175 *0.049+ 183 * 0.033+ 168 *0.029 (Accuracy = 1, R2 = 0.995, Q2 = 0.931, RMSE = 0.239, K = 10, AUC = 0.962 (95%CI: 0.878–1)) (Figs 7 and 8). Divided the above coefficient by the numbers of hydrogen atoms of the metabolite corresponding to the

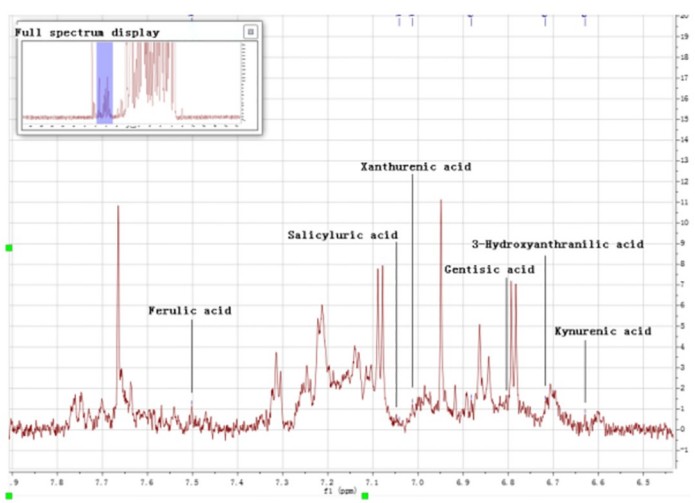

**Fig 7. The peaks of six principal components on the spectrum.**

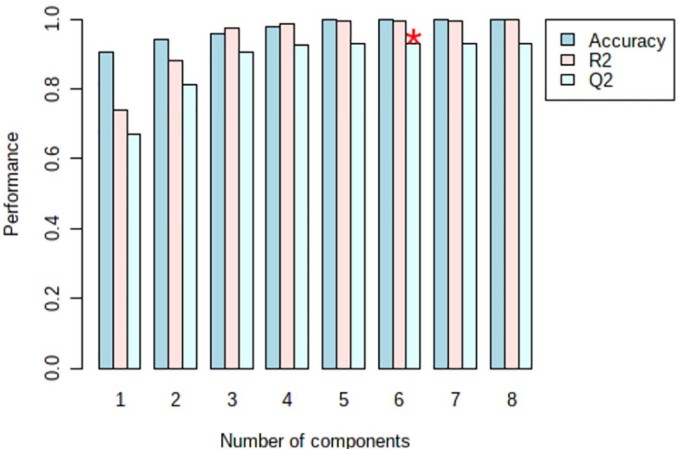

**Fig 8. PLS-DA classification.** The red star indicates the best classifier. (Accuracy = 1, R2 = 0.995, Q2 = 0.931.

interval to get the following model: Comp1 = XA $^*$0.033+-3-HAA $^*$0.024+ GA $^*$0.009+ SUA $^*$0.005+ FA $^*$ 0.003+ KA $^*$0.029. This model has the largest area under the curve (AUC) (0.932 (CI: 0.869–0.995)) compared with those six differential metabolites' peak intensity through univariate receiver operating characteristic curve (ROC) analysis [16] (Fig 9).

**4 Metabolic pathway analysis.** Though analyzing the above 15 differential metabolites' pathways by using pathway enrichment analysis and topological analysis in MetaboAnalyst 5.0 combined with Kyoto Encyclopedia of Genes and Genomics (KEGG), 9 metabolic pathways

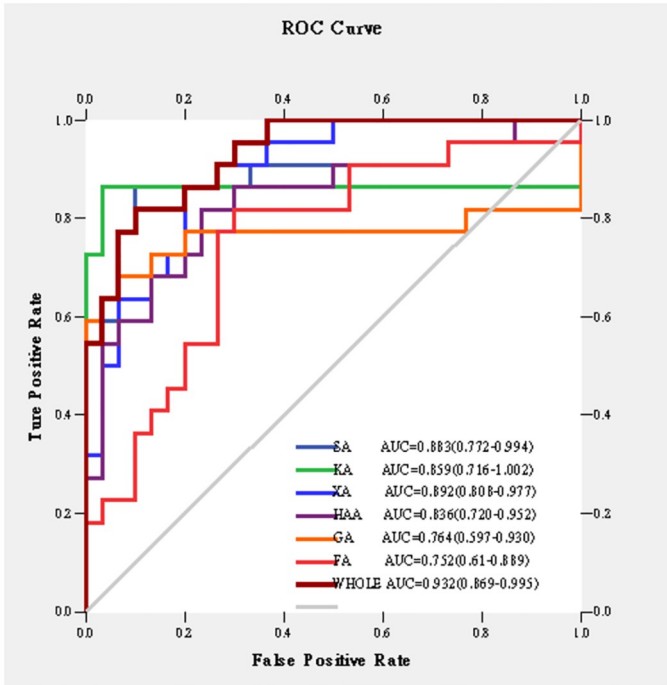

**Fig 9. Sensitivity comparison.** SA: Salicyluric acid; KA: Kynurenic acid; XA: Xanthurenic acid; HAA: 3-Hydroxyanthranilic acid; GA: Gentisic acid; FA: Ferulic acid; WHOLE: The prediction model, Comp1 = XA $^*$0.033 +3-HAA $^*$0.024+ GA $^*$0.009+ SUA $^*$0.005+ FA $^*$ 0.003+ KA $^*$0.004.

**Table 3. Result from pathway analysis by metaboanalyst5.0.**

| Pathway Name | Match Status | p | -log(p) | Holm p | FDR | Pathway Name |
|---|---|---|---|---|---|---|
| Tryptophan metabolism | 2/41 | 0.0047345 | 2.3247 | 0.04261 | 0.030557 | 0.19462 |
| Tyrosine metabolism | 2/42 | 0.00748 | 2.1261 | 0.05984 | 0.030557 | 0.00115 |
| Glutathione metabolism | 1/28 | 0.01289 | 1.8898 | 0.090228 | 0.030557 | 0.02698 |
| Phenylalanine metabolism | 1/10 | 0.016976 | 1.7702 | 0.10186 | 0.030557 | 0.2619 |
| Phenylalanine, tyrosine and tryptophan biosynthesis | 1/4 | 0.016976 | 1.7702 | 0.10186 | 0.030557 | 0 |
| Pyrimidine metabolism | 1/39 | 0.091057 | 1.0407 | 0.36423 | 0.13659 | 0.01726 |
| Aminoacyl-tRNA biosynthesis | 1/48 | 0.1078 | 0.9674 | 0.36423 | 0.13859 | 0 |
| Porphyrin and chlorophyll metabolism | 1/30 | 0.14884 | 0.82729 | 0.36423 | 0.16625 | 0.05288 |
| Riboflavin metabolism | 1/4 | 0.16625 | 0.77924 | 0.36423 | 0.16625 | 0 |

that may be affected between groups were obtained. There are significant differences in tryptophan metabolism, tyrosine metabolism, glutathione metabolism, phenylalanine metabolism and phenylalanine-tyrosine-tryptophan synthesis pathways between the two groups (all $P < 0.05$) Table 3. Moreover, the impact of the tryptophan metabolic pathway is greater than 0.1, and the number of accurate matching metabolites is 2. This metabolic pathway is considered to have the greatest impact on the metabolic pathway difference between the CBO group and the ABO control group (Fig 10). Then, the diagrams of different metabolic pathways and tryptophan metabolism and tyrosine metabolism pathways are drawn according to KEGG (Figs 11 and 12). Finally, the pairwise correlation of differential metabolites were analyzed by using Debiased Sparse Partial Correlation (DSPC), suggesting that GA and KA (P = 3.45e-07,

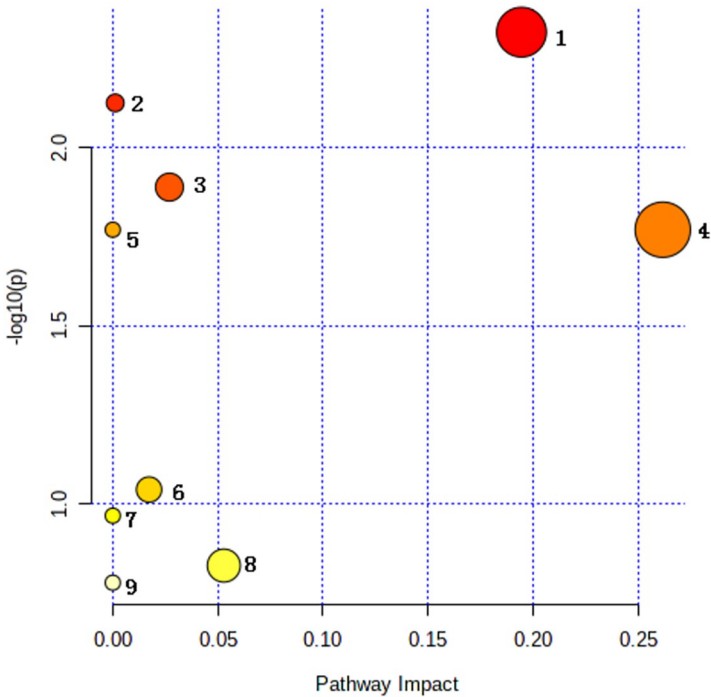

**Fig 10. The view of nine affected metabolic pathways between two groups.** 1 Tryptophan metabolism pathway, 2 Tyrosine metabolism pathway, 3 Glutathione metabolism pathway, 4 Phenylalanine metabolism, 5 Phenylalanine, tyrosine and tryptophan biosynthesis, 6 Pyrimidine metabolism pathway, 7 Aminoacyl-tRNA biosynthesis pathway, 8 Porphyrin and chlorophyll metabolism pathway, 9 Riboflavin metabolism pathway.

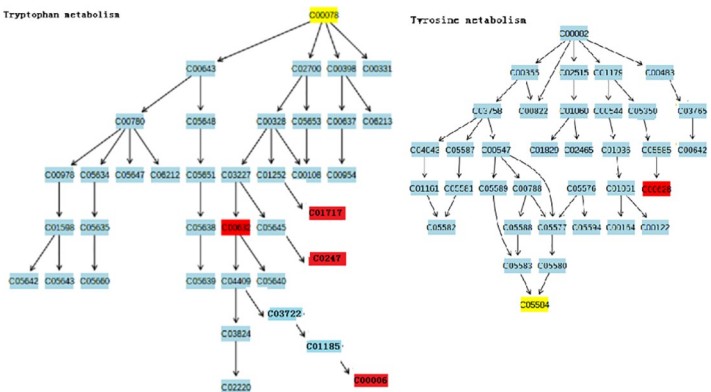

**Fig 11. The view of tryptophan and tyrosine metabolism pathway.** C00078 L-Tryptophan; C01717 Xanthurenate; C00632 3-Hydroxyanthranilic acid; C00247 Kynurenic acid; C00006 NADPH(Glutathione metabolism); C00628 Gentisic acid; C05584 Vanillylmandelic acid.

PACF = 0.84), GA and VMA (P = 7.7 e-05, PACF = 0.501), XA and CDP (P = 4.07e-06, PACF = 0.698) levels showed significant positive correlation, while there was negative correlation between L-Try and GA (P = 0.0016, PACF = -0.34), Try and XA (P = 0.0282, PACF = -0.227), KA and CDP (P = 0.0398, PACF = -0.285) levels.

## Discussion

The fast, untargeted NMR-metabolomic fingerprinting of biofluids has the ability to identify the individual metabolic phenotype and the signature of different diseases [17]. The difference in serum metabolomics between patients with colorectal cancer (CRC) who are not in a state of obstruction and healthy people is mostly reflected in the differences in glycolysis represented by pyruvate and lactic acid, and differences in glycerol and fatty acid metabolism represented by hydroxybutyric acid [4, 6, 7, 13, 15, 18, 19]. This study used 1H NMR to analyze the serum metabolites of patients with colon cancerous bowel obstruction (CBO) and patients

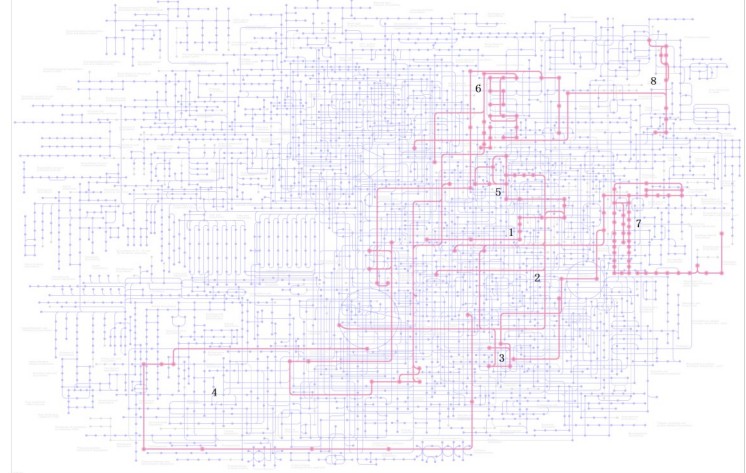

**Fig 12. The KEGG metabolism pathway map.** 1, Tryptophan metabolism 2, Tyrosine metabolism 3, Glutathione metabolism 4, Phenylalanine metabolism 5, Phenylalanine, tyrosine and tryptophan biosynthesis 6, Pyrimidine metabolism chlorophyll metabolism 7, Porphyrin and chlorophyll metabolism 8, Riboflavin metabolism.

with adhesive bowel obstruction (ABO) for the first time, and found that the metabolic differences mainly reflect in the metabolism of aromatic amino acids between the two groups. In particular, the up-regulation of tryptophan catabolism, which is based on the kynurenine pathway, may further decrease the cellular immune function and bowel smooth muscle's neurotransmitter conduction in CBO.

The pathological processes of bowel obstruction includes increasing secretions in the digestive tract and insufficient peripheral perfusion caused by the fluid accumulation of the third space, which leads to the decline of intestinal peristalsis, intestinal epithelial injury and inflammatory reactions. Intestinal secretions accumulate in the intestinal lumen, and when the reabsorption process fails, the intestine continues to lose fluid and electrolytes, leading to a vicious cycle of "expansion-secretion-expansion" [20], which directly leads to tissue glycolysis increasing and ischemia of Intestinal mucosa. Then there is a surge in lactic acid and fatty acids in the circulation [21]. Such metabolic changes also partially appears in the serum of non-obstructed CRC patients, which should be an important reason why the serum did not show significant differences in glycolysis and glycerol fatty acid metabolism pathways between CBO and ABO in this study. It's found that the difference in serum metabolism between the two groups is mainly reflected in the tryptophan, tyrosine and phenylalanine metabolism pathways (all P<0.05), especially the difference in the tryptophan metabolism pathway has the greatest impact (impact = 0.19). Tan et al. [18] observed differential metabolites of tryptophan, phenylalanine and tyrosine metabolism in the non-obstructed CRC serum. These metabolites are related to the co-metabolism of gut microbes and hosts. It is speculated that abnormal metabolism of intestinal flora may be an obvious metabolic feature of CRC. Continuous intestinal obstruction causes the necrosis of intestinal mucosal cells and the decline of intestinal smooth muscle function, which in turn causes a sharp decline in intestinal barrier function. The bacterial load in the intestine may exceed the host's defense capabilities, such as the obvious proliferation of Pseudomonas aeruginosa, Escherichia coli, Staphylococcus epidermidis, Candida and Enterococcus [22]. When obstruction occurs, CRC patients' fragile intestinal ecological balance is further destroyed, which may be the reason why the aromatic amino acid metabolism of CBO is more active than ABO.

In mammals, tryptophan (Try) catabolism is a physiological regulation to maintain immune homeostasis and immune tolerance, which avoids acute and chronic excessive inflammation and autoimmunity [23]. Colon cancer cells (such as HT-29, Caco-2 and LC-180 cells) have a complete Try catabolism mechanism, and its decomposition rate is several times higher than that of normal colonic epithelium [24]. Try is excessively consumed due to the high expression of indoleamine 2,3-dioxygenase (IDO-1/-2) In CRC patient [25]. According to the tryptophan depletion theory [26]: Try starvation may affect this the local utilization of amino acids which inhibits the proliferation of T cells and create an immunosuppressive environment that results in the body's tolerance to potentially immunogenic tumor antigens. When bowel obstructs, the intestinal barrier function is lost, and the enteric infection eventually causes systemic inflammatory syndrome, a large number of inflammatory mediators (such as IFN-G and IL-6) stimulate IDO1 excessive expression [27] makes Try catabolism more active in CBO patients. More than 95% of L-Try degradation produces L-kynurenine and a variety of downstream metabolites through the kynurenine pathway (KP), which are collectively referred to kynurenines (Kyns), including 3-Hydroxyanthranilic acid (3-HAA), Quinolinic acid, Xanthurenic acid (XA) and Kynurenic acid (KA), etc. [28]. In this study, the serum Try level of the CBO group was found significantly lower than that of the ABO group (t = -2.65, P = 0.011), while the CBO's levels of XA (t = 10.43, P = 3.87E-14), 3-HAA (t = 9.1236, P = 3.21E-12) and KA (t = 4.5773, P = 3.14E-05) were significantly higher than ABO's. Malignant tumors are prone to cause malnutrition due to their strong metabolism in the process of

growth. When patients are suffering from bowel obstruction, the degree of malnutrition and cachexia increases significantly. When CRC patient suffers from inflammatory reaction caused by obstruction, the higher ratio of Kyns/Try that caused by rapid consumption of Try and the excessive production of Kyns is a possible disease identification index. Kyns/Try ratio increasing may indicate tumorigenesis [29] and ischemic necrosis of muscle cells [30]. It's found that CBO's serum L-Try level is negatively correlated with XA level (P = 0.0282, PACF = -0.227).

XA has a competitive inhibitory effect on the transport of L-glutamate (L-Glu) [31]. By blocking the transport of L-Glu to synaptic vesicles, it ultimately reduces the release of L-Glu from synapses, thereby glutamatergic transmission is inhibit [30]. KA is an antagonist of the Glu receptor (GluR). The overexpression of KA in the nerve injury model can block the conduction of Glu [32]. Glu is the main excitatory neurotransmitter in the central nervous system [33], and it may also act as an excitatory neurotransmitter in the enteric nervous system. The excitatory transmission of intestinal smooth muscle is mainly cholinergic in nature. Glu activates N-methyl-D-aspartic acid receptors (NRs) and promotes the release of acetylcholine of neurons in the ileum and colon [34]. Therefore, the peristalsis may be affect by the Kyns' inhibition to the Glu transport and GluR of enteric neurons. When CRC patients suffer from obstruction, the level of Kyns increases significantly, which means that peristalsis is further weakened.

KA has also been identified as an endogenous aryl hydrocarbon receptor (AhR) agonist [35]. In addition to impairing immune T cell function through tryptophan starvation, high levels of KA produced by DCs and tumors under inflammatory conditions caused by bowel obstruction can activate IL-10 in DCs and NK cells though AhR activating [36] and IL6 transcription in cancer cells and macrophages [37] in the absence of Trp, which causes immunosuppression and immune tolerance. 3-Hydroxyanthranilic acid (3-HAA) is an intermediate metabolite and precursor of the excitotoxin Quinolinic acid with high redox activity [38]. It inhibits T cells' expression and increases the percentage of Tregs by inducing the expression of heme oxygenase 1 (HMOX-1) [39], consuming intracellular glutathione, inhibiting cytokine release [40], inducing T cells that produce antigen-specific IL-10 [41] and inhibiting PDK1 expression [42], therefor the immune response is directly damaged. Compared with ABO patients, the high level of Kyns in the serum of CBO patients means more severe immunosuppression and immune escape occuring.

Gentisic acid (GA) is usually considered to be a salicylic acid's metabolite catalyzed by CYP450 enzyme in the liver, but endogenous GA is produced by oxidation of Homogentisic acid, a common metabolite of phenylalanine and tyrosine. Then GA produces Maleylpyruvic acid which enters the pyruvate metabolism. Vanillylmandelic acid(VMA) is also a downstream metabolite of tyrosine. In this study, it 's found that the levels of GA and VMA have a higher correlation (P = 7.7e-05, PACF = 0.501), which proves that the tyrosine metabolism of CBO is more active than that of ABO. GA inhibits the activity of cdK1 enzyme in vitro and has a highly inhibitory effect on the cell proliferation of three different colon cancer cell lines (HcT-116, HT-29 and Mda-MB-231) [43]. Salicyluric acid (SUA) is formed by the conjugation of salicylic acid (SA) and glycine [44], and SUA is the main metabolite of SA. Endogenous SA is formed by oxidation of benzoic acid, a downstream product of phenylalanine metabolism, mediated by NADPH. SUA has a DNA repairing function, because it has a phenolic hydroxyl structure that similar to tannic acid, which has an inhibitory effect on the genotoxicity induced by mitomycin C (MMC) [45]. Endogenous Ferulic acid (FA) is produced by caffeic acid, the downstream product of phenylalanine metabolism, under the action of methyltransferase. It's an antioxidant that can reduce the strength of various inflammatory mediators. At the same time, its phenolic core and expanded side chain conjugationt structure make it have anti-cancer properties. It interferes the signal pathways that control cell growth in cells, and activates programmed cell death and oxidative stress-related responses [46]. Resveratrol combined with

Ferulic acid can inhibit 3D proliferation in vitro by increasing the expression of p15 in colon cancer HCT116 cells [47]. It's found that the serum levels of GA (t = 6.8674, P = 9.68E-09), SUA (t = 6.2593, P = 8.64E-08) and FA (t = 4.9326, P = 9.38E-06) of CBO patients are significantly higher than those of ABO patients. These three metabolites that belong to phenylalanine and tyrosine metabolism pathways can exert anti-cancer effects through different mechanisms in different cell types. Therefore, whether up-regulating the phenylalanine and tyrosine metabolism pathways can inhibit the growth of colon cancer cells requires further research.

## Conclusions

This study confirms that the serum metabolic differences are mainly reflected in the tryptophan catabolism between the CBO and ABO by 1H-NMR analysis. Serum metabolomics can distinguish CBO from ABO, and the entire NMR spectrum can be used as a collective "biomarker", which convinces us that there is a feasibility to develop a new serum metabolites tool for CBO diagnosis. The nuclear overhauser effect spectroscopy (NOESY) assay was not performed in this research, so there should be flaws in the identification of lipoproteins [48], but our original purpose is to conduct small molecule observations, and the analysis of macromolecular proteins will be further discussed in future experiments. Meanwhile, the sample size of this research is very small. In order to discover highly specific and sensitive metabolic markers that can help clinicians making rapid decision, the comprehensive application of various metabonomics techniques(GC-MS, LC-MS), the unification of sample collection standards and the verification of larger populations are necessary.

## Supporting information

**S1 Data.**
(DOCX)

## Author Contributions

**Conceptualization:** Jie Zeng, Hua Jiang.

**Data curation:** Jin Peng, Pengchi Deng, Kexun Li, Kai Wang.

**Funding acquisition:** Jie Zeng.

**Investigation:** Jie Zeng, Kexun Li, Daolin Long.

**Methodology:** Jin Peng.

**Project administration:** Jie Zeng.

**Software:** Jin Peng, Pengchi Deng, Kai Wang.

**Supervision:** Hua Jiang.

**Visualization:** Jin Peng.

**Writing – original draft:** Jie Zeng.

**Writing – review & editing:** Jie Zeng, Jin Peng.

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
