## [Decision Letter · Decision Letter 0]

24 Nov 2021

PONE-D-21-25026

Establishment of an early diagnosis model of colon cancerous bowel obstruction based on 1HNMR

PLOS ONE

Dear Dr. Peng,

Thank you for submitting your manuscript to PLOS ONE. After careful consideration, we feel that it has merit but does not fully meet PLOS ONE’s publication criteria as it currently stands. Therefore, we invite you to submit a revised version of the manuscript that addresses the points raised during the review process.

The paper was considered interesting by the reviewer, but a number of issues must be addressed before we can further consider the work.

We look forward to receiving your revised manuscript.

Kind regards,

Oscar Millet

Academic Editor

PLOS ONE

A clean copy of the edited manuscript (uploaded as the new *manuscript* file).

“The present study was supported by Basic scientific research business expenses of public welfare scientific research institutes of Sichuan Province, China (grant no. 30504010428).”

“No, The funders had no role in study design, data collection and analysis, decision to publish, or preparation of the manuscript”

6. Thank you for stating the following in your Competing Interests section: 

“NO authors have competing interests”

Reviewers' comments:

Reviewer's Responses to Questions

**Comments to the Author**

1. Is the manuscript technically sound, and do the data support the conclusions?

Reviewer #1: Partly

2. Has the statistical analysis been performed appropriately and rigorously? 

Reviewer #1: Yes

3. Have the authors made all data underlying the findings in their manuscript fully available?

Reviewer #1: No

4. Is the manuscript presented in an intelligible fashion and written in standard English?

Reviewer #1: Yes

5. Review Comments to the Author

Reviewer #1: In this manuscript 1H NMR coupled with machine learning is used to build a model for the differential diagnosis of acute colon cancerous bowel obstruction (CBO) vs. adhesive bowel obstruction (ABO) in serum. Although the entire spectrum is proposed to function as “biomarker”, then the authors concentrate the discussion on six metabolites.

It would be important to know what would be the prediction accuracy, sensitivity and specificity of a model based on these metabolites rather than on the whole binned spectrum, to evaluate the contribution of the other molecules.

The NMR analysis was limited to the acquisition of CPMG spectra, thus excluding all lipoproteins components, which would help defining metabolic alterations. It is a common practice in 1H NMR metabolomics to use at least NOESY and CPMG experiments.

The serum handling procedure are largely divergent with respect to ISO standards (ISO 23118:2021 Molecular in vitro diagnostic examinations — Specifications for pre-examination processes in metabolomics in urine, venous blood serum and plasma), which could in principle affect the outcome of the downstream metabolomics analysis. Possibly the adopted procedure does not influence the comparison between the two groups of patients enrolled in the same study, but it might affect the general applicability of the model.

It would be also important to know what has been administered to the patients in the time interval between admission and blood collection (e.g. Ringer acetate solutions or contrast agents for imaging, and not just “drugs” affect the NMR metabolic profiles, Metabolomics 2015;11(6):1769-1778.).

Formal aspects:

Missing information: the data availability statement does not describe where the data can be found

Spelling:

Throughout the text: 1H NMR

Page 10, … 5 seconds

Page 12 (Discussion), … between the two groups

6. PLOS authors have the option to publish the peer review history of their article (what does this mean?). If published, this will include your full peer review and any attached files.

Reviewer #1: No

---

## [Author Response · Author response to Decision Letter 0]

23 Dec 2021

Response to academic editor：

We really appreciate you for your carefulness and conscientiousness. Your suggestions are really valuable and helpful for revising and improving our paper. According to your suggestions, we have made the following revisions on this manuscript:

1. Please ensure that your manuscript meets PLOS ONE's style requirements, including those for file naming. The PLOS ONE style templates can be found athttps://journals.plos.org/plosone/s/file?id=wjVg/PLOSOne_formatting_sample_main_body.pdf and https://journals.plos.org/plosone/s/file?id=ba62/PLOSOne_formatting_sample_title_authors_affiliations.pdf.

Response：Thank you for your detailed comments. We have revised the manuscript according to the PLOSOne guideline thoroughly, which are highlighted in red in the revised manuscript.

Response：Thank you very much for your advice. The title page have been added into the beginning which are highlighted in red in the revised manuscript. (page 1, lines5-12)

3. We suggest you thoroughly copyedit your manuscript for language usage, spelling, and grammar. 

A clean copy of the edited manuscript (uploaded as the new *manuscript* file).

Response：Thank you for your detailed comments. We have thoroughly copyedited the manuscript for language usage, spelling, and grammar. The amendments are highlighted in red in the revised manuscript.( page 1, lines1,30,38; page3, line42; page5, lines16,18-20; page 6, lines9,12,25-26; page 7, lines11-12; page 9, lines6-7.)

Response：Thank you for your detailed comments. The correct grant numbers is 2018YSKY0017-9( Basic scientific research business expenses of Science ＆ Technology Department of Sichuan Province) , which is also corrected in the ‘Funding Information’ section.

5. Thank you for stating the following in the Acknowledgments Section of your manuscript:“The present study was supported by Basic scientific research business expenses of public welfare scientific research institutes of Sichuan Province, China (grant no. 30504010428).”

“No, The funders had no role in study design, data collection and analysis, decision to publish, or preparation of the manuscript”. Please include your amended statements within your cover letter; we will change the online submission form on your behalf.

Response：Thank you for your positive comments. We have removed funding-related text from the manuscript. The study was supported by Basic scientific research business expenses of Science ＆ Technology Department of Sichuan Province (grant no.2018YSKY0017-9), and the funders had no role in study design, data collection and analysis, decision to publish, or preparation of the manuscript. Kexun Li and Daolin Long who were studying for a master's degree got ￥12,000 as Labor wage from this foundation, respectively. Mean while, the other authors of this manuscript got no salary from the funder.

6. Thank you for stating the following in your Competing Interests section: “NO authors have competing interests”

Response：Thank you for your detailed comments. The authors have declared that no competing interests exist, which are highlighted in red in the revised manuscript. (page 9, lines 29-30)

Response： The full ethics statement has been included in the ‘Methods’ section of the manuscript file, which is highlighted in red in the revised manuscript.( page 2, lines 19)

Response to reviewer #1: 

We really appreciate you for your carefulness and conscientiousness. Your suggestions are really meaningful and useful for revising and improving this paper. According to your suggestions, we have made the following revisions on this manuscript:

1. In this manuscript 1H NMR coupled with machine learning is used to build a model for the differential diagnosis of acute colon cancerous bowel obstruction (CBO) vs. adhesive bowel obstruction (ABO) in serum. Although the entire spectrum is proposed to function as “biomarker”, then the authors concentrate the discussion on six metabolites.

It would be important to know what would be the prediction accuracy, sensitivity and specificity of a model based on these metabolites rather than on the whole binned spectrum, to evaluate the contribution of the other molecules.

Response：We appreciate the reviewers' feedback. Regarding the accuracy, sensitivity, and specificity of the model composed of six metabolites in comparison to the entire classification spectrum for predicting CBO, we searched the HMDB for 1470 possible metabolites after obtaining 35 segments of chemical shifts with VIP values greater than 1 via PCA and PLS-DA analysis of the spectral matrix, of which 388 possible endogenous metabolites were detected in blood. According to the Jaccard value, the top 30 metabolites were selected. By comparing standard spectra with HMDB (1H NMR Spectrum [1D, 600 MHz, D2O, anticipated]), it was shown that 15 metabolites had the lowest peaks at chemical shifts that overlapped with the above 35 segments. Mnova software was used to determine the peak intensities of these 15 shifts separately, and a t-test revealed that the peak intensities of six shifts were substantially different. Six chemical shifts correspond to six metabolites, and the integrals of the lowest peak intensity are notably different S2 Table. We have revised the article in light of the reviewers' comments. (page 5, lines 2-28)

2. The NMR analysis was limited to the acquisition of CPMG spectra, thus excluding all lipoproteins components, which would help defining metabolic alterations. It is a common practice in 1H NMR metabolomics to use at least NOESY and CPMG experiments.

Response：We appreciate your kind words. Due to financial and time constraints, we have only completed the 1H NMR test at this stage. We will use GC-MS or LC-MS for the mass spectrometry test verification of the above substances in the next stage. We have also revised the article in light of the reviewers' comments. (page 8, line 41)

3. The serum handling procedure are largely divergent with respect to ISO standards (ISO 23118:2021 Molecular in vitro diagnostic examinations — Specifications for pre-examination processes in metabolomics in urine, venous blood serum and plasma), which could in principle affect the outcome of the downstream metabolomics analysis. Possibly the adopted procedure does not influence the comparison between the two groups of patients enrolled in the same study, but it might affect the general applicability of the model.

Response： Regarding the specimen handling process, we made an error expression: Because of the particularity of emergency patients (more admitted at night), our plan was to draw whole blood from the peripheral veins of all participants within 2 hours after admission, and used vacuum blood collection tubes to collect 3ml of whole blood for each participant (vacuum tube with blue cap without addition, 10.25mm×64mm, batch number 363095, American BD company), immediately put it in a refrigerator at -20℃, and centrifuge at 3000r/min for 15min within 48h, and took the supernatant , Transferred to EP tube, -80 ℃ refrigerator for refrigeration. But the actual operation was to draw 3ml of whole blood from the peripheral vein within 2 hours after admission into the tubes (Blue cap with sodium citrate, 10.25mm×64mm, batch number 363095, US BD company), centrifuged inside at 16000r/min for 15min in 30 minutes, took the supernatant, transferred to EP tube, and refrigerate at -80℃. After the specimens were collected, they were transferred to the laboratory to thaw at room temperature and transferred to a 5mm Wilmad NMR tube for on-board testing and analysis. Although the thawing temperature is different from that specified in ISO 23118:2021 and requires extended time detection, we believe that, while the serum handling procedures are different, the differences determined by the method based on the overall molecular magnetic resonance hydrogen spectroscopy profile are similar. That is, our method, PLS-DA, PCA, relies heavily on linear transformations to identify the components with the highest variation among the different grouped samples, and so is mostly unaffected by the aforementioned criteria. We have also revised the article in light of the reviewers' comments. (page 3, lines 22-29, 32-33; )

4. It would be also important to know what has been administered to the patients in the time interval between admission and blood collection (e.g. Ringer acetate solutions or contrast agents for imaging, and not just “drugs” affect the NMR metabolic profiles

Response：Thank you very much for your advice. Within 48 hours, none of our patients had medical procedures (e.g., oral or intravenous administration, gastrointestinal decompression, enemas), with the exception of fasting and water fasting for >24 hours, which we included in our exclusion criteria. No patient received imaging examination using contrast agents, and no patient got intravenous or oral rehydration prior to specimen collection. We have already revised the article in light of the reviewers' comments. (page 2, lines 38,44; page 3, lines 1-3 )

5. Formal aspects:

Missing information: the data availability statement does not describe where the data can be found.

Response：Thank you for your constructive comments. In accordance with the principles of medical ethics, the raw data of the desensitized papers can be downloaded at https://github.com/dcpengjin/metabolomics_data.git

6. Spelling:

Throughout the text: 1H NMR 

Page 10, … 5 seconds

Page 12 (Discussion), … between the two groups.

Response： We appreciate the reviewers' helpful and constructive suggestions for grammar and spelling changes. All errors mentioned above have been corrected. We have already revised the article in light of the reviewers' comments. (page 1, lines 2,15; page 3, lines 41-42; page 6, lines 9,12,26)

---

## [Decision Letter · Decision Letter 1]

28 Feb 2022

PONE-D-21-25026R1Establishment of an early diagnosis model of colon cancerous bowel obstruction based on 1HNMRPLOS ONE

Dear Dr. Peng,

Thank you for submitting your manuscript to PLOS ONE. After careful consideration, we feel that it has merit but does not fully meet PLOS ONE’s publication criteria as it currently stands. Therefore, we invite you to submit a revised version of the manuscript that addresses the points raised during the review process. Specifically, there are still a couple of points raised by the reviewer that require to be amended.

We look forward to receiving your revised manuscript.

Kind regards,

Oscar Millet

Academic Editor

PLOS ONE

Journal Requirements:

Reviewers' comments:

Reviewer's Responses to Questions

**Comments to the Author**

1. If the authors have adequately addressed your comments raised in a previous round of review and you feel that this manuscript is now acceptable for publication, you may indicate that here to bypass the “Comments to the Author” section, enter your conflict of interest statement in the “Confidential to Editor” section, and submit your "Accept" recommendation.

Reviewer #1: (No Response)

2. Is the manuscript technically sound, and do the data support the conclusions?

Reviewer #1: Partly

3. Has the statistical analysis been performed appropriately and rigorously? 

Reviewer #1: Yes

4. Have the authors made all data underlying the findings in their manuscript fully available?

Reviewer #1: Yes

5. Is the manuscript presented in an intelligible fashion and written in standard English?

Reviewer #1: Yes

6. Review Comments to the Author

Reviewer #1: I believe the previous comments 1 and 2 have been misunderstood.

1. The question was very simple: what is the difference in prediction accuracy, sensitivity and specificity if the authors use the entire binned spectrum? This procedure does not require any assignment (for details see for example https://doi.org/10.1016/j.trac.2018.10.036). I invite the authors to test this approach.

2. I was suggesting to use 1H NMR NOESY spectra to derive information on the lipoprotein components (see Bruker IVDR tool; https://doi.org/10.1021/acs.analchem.8b02412), not to perform MS analyses.

7. PLOS authors have the option to publish the peer review history of their article (what does this mean?). If published, this will include your full peer review and any attached files.

Reviewer #1: No

---

## [Author Response · Author response to Decision Letter 1]

19 Mar 2022

Response to academic editor：

Your suggestions are really valuable and helpful for revising and improving our paper. According to your suggestions, we have made the following revisions on this manuscript:

Response：Thank you for your detailed comments. We have reviewed our reference list thoroughly to ensure that it is complete and correct. we did not find any records about being retracted of these papers that we cited in the manuscript. In this revision, we cited another three papers.The amendments are highlighted in red in the revised manuscript.( lines195,216,339,424-427,498-500.)

Response to reviewer #1: 

Your suggestions are really meaningful and useful for revising and improving this paper. According to your suggestions, we have made the following revisions on this manuscript:

1. The question was very simple: what is the difference in prediction accuracy, sensitivity and specificity if the authors use the entire binned spectrum? This procedure does not require any assignment (for details see for example https://doi.org/10.1016/j.trac.2018.10.036). I invite the authors to test this approach.

Response： The papers you recommended are very valuable to us. We performed ROC analysis on the 6 metabolites which have significant differences between the two groups and the full-spectrum prediction model. According to the AUC value, the prediction accuracy of the full-spectrum model was higher. For the verified source code see: https://github.com/dcpengjin/metabolomics_data.git. We also have revised the article in light of the reviewers' comments. (lines 192-195)

2. I was suggesting to use 1H NMR NOESY spectra to derive information on the lipoprotein components (see Bruker IVDR tool; https://doi.org/10.1021/acs.analchem.8b02412), not to perform MS analyses.

Response：Thank you very much for your professional advice and reminders. Our original purpose was to observe small molecules through a relatively faster 1H-NMR test. Due to the economic and time constraints, this study did not carry out NOESY test, so it was flawed in identifying lipoproteins, and the analysis of macromolecular proteins will be further discussed in future experiments. We have also revised the article in light of the reviewers' comments. (line 337-341)

---

## [Editor Report · Decision Letter 2]

28 Mar 2022

Establishment of an early diagnosis model of colon cancerous bowel obstruction based on 1HNMR

PONE-D-21-25026R2

Dear Dr. Peng,

We’re pleased to inform you that your manuscript has been judged scientifically suitable for publication and will be formally accepted for publication once it meets all outstanding technical requirements.

Kind regards,

Oscar Millet

Academic Editor

PLOS ONE
---

## [Editor Report · Acceptance letter]

6 Apr 2022

PONE-D-21-25026R2 

Establishment of an early diagnosis model of colon cancerous bowel obstruction based on 1H NMR 

Dear Dr. Peng:

I'm pleased to inform you that your manuscript has been deemed suitable for publication in PLOS ONE. Congratulations! Your manuscript is now with our production department. 

Kind regards, 

on behalf of

Dr. Oscar Millet 

Academic Editor

PLOS ONE